# An Exploratory Investigation of the Effect of a Sports Vision Program on Grade 4 and 5 Female Netball Players’ Visual Skills

**DOI:** 10.3390/ijerph19169864

**Published:** 2022-08-10

**Authors:** Dané Coetzee, Elna de Waal

**Affiliations:** 1Physical Activity, Sport and Recreation (PhASRec), Focus Area, Faculty of Health Science, School of Human Movement Sciences, North-West University, Potchefstroom Campus, Potchefstroom 2531, South Africa; 2Department of Exercise and Sport Science, Faculty of Health Sciences, University of the Free State, Bloemfontein 9301, South Africa

**Keywords:** depth perception, hand–eye coordination, netball, ocular motor control, sports vision, visual skills

## Abstract

Vision is one of the most complex and dominant sensory systems necessary for information feedback from the environment. Few studies have already reported a positive effect of a sport vision program on elite sport teams’ visual skills; however, few studies have focused on the effect of sport vision programs on children’s visual skills. Therefore, the aim of this study was to determine the effect of a sports vision program on Grade 4 and 5 female netball players’ visual skills. Girls (N = 25) with a mean age of 10.08 years (SD = 0.65) formed part of this study. A pre-test–post-test design was followed with a retention test. The eight-week sports vision program was executed twice a week for 60 min on the experimental group (*n* = 13). The Developmental Test of Visual–Motor Integration (VMI-4), the Wayne Saccadic Fixator (WSF) and the Developmental Eye Movement (DEM) test were used to evaluate the girls’ visual skills, hand–eye coordination, visual reaction time, peripheral vision and saccadic eye movements. No statistical differences were found between the two groups before starting with the sports vision program. After intervention, significant differences between the two groups were reported, with the experimental group performing better in hand–eye coordination (*p* = 0.001) and reaction time (*p* = 0.001). Results further indicated that the experimental group experienced significant improvements (*p* ≤ 0.05) in motor coordination, hand–eye coordination, reaction speed and visual tracking after intervention with significant lasting effects (*p* ≤ 0.05). The control group performed significantly worse in reaction time (*p* = 0.01). A sports vision program can be recommended for Grade 4 and 5 female netball players to improve certain visual skills.

## 1. Introduction

Vision is one of the most complex and dominant sensory systems necessary for information feedback from the environment. Good vision requires good visual and/or perceptual skills, which includes the information that is sent from the eyes to the brain where information is interpreted [1]. Of all the skills that are required in sports, vision is the last skill to develop fully and the first to diminish during sports participation [2].

The visual system plays an important role in the development and improvement of motor performance of sportsmen [3,4,5] where approximately 80% of all sensory input to the brain originates from the eyes. Furthermore, the visual system is directly related to the proprioceptive system of the brain and is in close cooperation with the latter during exercise and sports-specific actions [6]. Calder [7] is of the opinion that the visual system undergoes strain during the execution of athletic activities while underdeveloped, thus negatively influencing the ability of the sportsperson to execute a specific skill in his/her sport. It seems that vision is an important factor in sports execution [7,8,9,10], and high-level sports performance can only be attained when both visual information and cognitive abilities are adequate [11].

According to Krüger et al. [12] and Wilson and Falkel [6], vision is both a learned and a developing skill because a person’s ocular motor control (eye muscle functions) can be trained, just like any other muscle, to function better. An effective visual system, which includes good eye functions for binocular alignment, convergence–divergence, fixation, visual tracking, visual memory, visual consecutive memory, peripheral vision and depth perception, is necessary because it contributes to the development of normal motor and sports skills [3,6,13]. Furthermore, vision and proprioception are important sensory modalities for the correct execution of motor skills and are the primary sources which receive information from the environment and determine how the brain and body should respond to this information [3,13,14,15,16].

Ocular motor control is regulated by three pairs of extraocular muscle (superior, inferior, medial and lateral rectus muscles, as well as the superior and inferior oblique), which make fixation, convergence, accommodation and visual tracking skills possible [3,17,18,19]. These muscles, which are found at the exterior of each eye, connect above, below and on either side of the sclera. They should be able to cooperate and work in balance, in order to provide coordinated and smooth eye movements which will ensure that the image that reaches each eye blends into one image in the visual cortex [3,6,13]. The eye muscles are also responsible for the eye movement, which helps to focus or fixate on and follow objects, in addition to ensuring simultaneous movement [3,6]. The coordination of the extraocular muscles is critical for effective and accurate eye movement [6]. Calder [7] states that the correct technical execution of skills during sports is a very important factor. Hopfinger et al. [20] confirms this statement by verifying that visual information is critical when a player’s movement must coincide with a changing sports environment, especially during motor skills where hand–eye and foot–eye coordination play a role [3]. According to Wilson and Falkel [6] hand–eye coordination, balance and visual skills (accommodation, visual tracking of teammates and opponents and peripheral awareness) are seen as the most important skills in basketball. From the literature, there appears to be a close relationship between basketball and netball, which is why the same visual and motor skills important in basketball (as stated by Wilson and Falkel, [6]) can be deduced to be important in netball.

Sports require high levels of motor development, perceptual skills (decision making), visual skills (eye movements) and hand–eye and foot–eye coordination [7]. Without good visual information, well-developed cognitive processing of the received visual information will not imply optimal sport performance [11]. If any faulty input of information occurs as a result of the visual system, the child’s reaction to information will consequently be faulty and could lead to motor deficits later on [13]. Pienaar [13] is of the opinion that perceptual skills such as laterality, direction, spatial orientation, tactile awareness, figure–ground development and visual perception are increasingly seen as determining factors for the execution of sports-specific actions which include, amongst others, coordination, balance, aerobic capacity, endurance, strength, muscle tone and flexibility. This research further indicates that a variety of ocular motor control skills (accommodation, fixation and visual tracking), stereopsis (depth perception), visual memory and visual consecutive memory are important visual skills in a sport match. A recent review also confirms that similarly to motor skills, visual skills also influence athletic performance [10]. This review by Presta et al. [10] provides evidence stipulating the importance of the sound stereopsis ability for sports participation, with specific reference to interception actions in ball sports. Strategic sports on the other hand (such as soccer and hockey) mainly requires peripheral and spatial vision [10], perception span and spatial working memory, while utilization of these visual skills can differ depending on playing position and field measures of the relevant sport [21].

The literature indicates that a sports vision program has a positive effect on the improvement of visual skills such as ocular motor control (visual tracking, convergence–divergence, and fixation), peripheral vision, depth perception and hand–eye coordination, during different sports codes which include soccer, rugby, cricket, baseball, hockey and racket sports [6,10,12,22,23,24,25]. Sports vision can now be defined as the development and improvement in peripheral vision, visual concentration, hand–eye coordination, ocular motor control and reaction time [23]. Wilson and Falkel [6] and Krüger et al. [12] verify the prior mentioned studies by stating that a sports vision program further increases the efficacy of the visual system and then contributes to the improvement in the sportsperson’s motor abilities as well as their sports performance. The development and improvement of visuomotor skills in sport are not only seen on a behavioral level, but also at a central nervous system level where neural function modulations as well as grey and white matter structure changes are reported adaptations [26]. A recent review by Appelbaum and Erickson [27] confirms that sport vision training improves sport performance by means of enhancing visual perception, cognitive skills and/or oculomotor skills. In addition, the new digital age also plays an important role in aiding sports vision programs by means of virtual reality simulations and perceptual learning-inspired training programs [27].

Research by Du Toit et al. [24], on 20 female rugby players between the ages of 19 and 24, verifies the positive effect of a sports vision program on visual skills and hand–eye coordination. In addition to the prior mentioned study, Du Toit et al. [28] found that the hand–eye coordination of schoolboys between the ages of 14 and 17 improved after following a sports vision program over a period of five weeks. Krüger et al. [12] presented a sports vision program for eight weeks, twice a week, for 60 min to 13 provincial under-19 cricket players. The findings of this study indicated that the ocular motor control (fixation and visual tracking), visual awareness and visual perception of the players who took part in the sports vision program had improved. Although research indicates prevalent positive effects following a sports vision program, the research findings by Abernethy and Wood [23] show contradictory results. These researchers report no significant improvement in the visual skills of 40 male and female participants (16 to 28 years of age), of which none had experience in racket sports, after a sports vision program had been followed for four weeks, 20 min, four times a week [20]. The findings by Abernethy and Wood [23] might be attributed to the fact that the sports vision program that had been followed focused on general visual skills and not on sport-specific visual skills.

Upon following a comprehensive literature search, it has become apparent that information with regards to a sports vision program for the visual skills of young netball players is inadequate, consequently pointing out a gap in this regard. Accordingly, the research question that needs to be answered with this study is whether a sports vision program will improve the visual skills (ocular motor control, depth perception, visual–motor integration, visual perception, motor coordination and reaction speed) and hand–eye coordination of Grade 4 and 5 netball players.

## 2. Materials and Methods

### 2.1. Study Design

A pre-test–post-test followed by a retention test (2 months later) was used during this study to do an exploratory investigation of the effect of the sports vision program on the visual skills of Grade 4 and 5 netball players.

### 2.2. Participants

The research population used for this study was Grade 4 and 5 girls. An availability sample of 25 girls (*n* = 13, Grade 4; and *n* = 12, Grade 5), identified by teachers as netball players in the school, was selected based on availability from one primary school. Only participants who had written parental permission and had given assent themselves, formed part of this study. Furthermore, if the participant made use of any visual aid (for example glasses or contact lenses) to correct their vision or were diagnosed with any other visual condition (such as strabismus or diplopia), or participated in any visual stimulation intervention program, they were excluded from the study. After data collection, the participants were randomly divided into an experimental group (*n* = 13) and a control group (*n* = 12). During the post-test and retention test opportunities, one test subject in the control group did not take part in the tests because of medical reasons, whose data were consequently left out of this study. See Figure 1 for a detailed description.

### 2.3. Measuring Instruments

#### 2.3.1. Wayne Saccadic Fixator (WSF)

The Wayne Saccadic Fixator (WSF) is a flat, adjustable panel of 1 m × 1.2 m, which is mounted against a wall at eye level and was done with both eyes open. The test subject stands at his/her own arm’s length from the middle of the panel and focuses on the middle of the WSF [29]. The instrument is controlled by a computer disc, generating a variation of light patterns, to which the test subject reacts by selecting the flashing lights. During the execution of the first test (Procedure 1: pro-action), hand–eye coordination is determined [29]. During Procedure 1 a combination of 33 red lights are involuntarily flashed for a period of 30 s. The lights go on one by one, and the light that goes on first remains on until the test subject touches it with the tips of their fingers. This flashing light then goes off and the next light starts to flash immediately. This continues for 30 s, and the total amount of lights touched in this period is noted as the score. The second test (Procedure 21: reaction), assesses the test subject’s hand–eye coordination, peripheral vision, and reaction time. This test consists of a 30 s evaluation during which the lights flash randomly and the test subject must react upon them by pressing on the light before it goes off [30]. The third test (Procedure 18) assesses reaction speed [29]. This test begins with a light going on at the nine o’clock mark. The test subject controls when the test starts by pressing on this light with their fingertips, after which the nine o’clock light goes off and the three o’clock light immediately goes on. The test subject must press the light on the three o’clock mark as fast as possible with the same hand; the time it takes the test subject to press the two lights is noted. The distance between the nine o’clock mark and the three o’clock mark is 71 cm. According to Vogel and Hale [31], the validity for Procedure 1 and 21 is r = 0.82 and Procedure 18 is r = 0.69.

#### 2.3.2. The Developmental Eye Movement (DEM) Test

The Developmental Eye Movement (DEM) test is a clinical test which determines saccadic eye movements (visual tracking) with children. Administering the DEM has two vertical series, namely Test A and Test B (read from top to bottom), each consisting of forty single digit numbers that the child should be able to read out loud. The numbers are evenly spaced into two vertical columns. The measure of the vertical series (Test A and Test B) is determined by the time that it takes to read the numbers in Test A and Test B from top to bottom, respectively. The final time of the vertical measure is determined by adding together the respective times that it took to complete Test A and Test B. Then this is followed by a horizontal series, namely, Test C (read from left to right), which consists of 15 rows containing five numbers per row, but these numbers are unevenly distributed in each row. This is also a timed test [32]. The horizontal measure is executed by capturing the time it takes the person to complete Test C. Any errors that occur during the readouts of the numbers by name could be an indication of a disturbance in the visual system, or an inadequacy to recognize numbers. Any replacement, omission, addition and displacement mistakes must be noted. After the times have been processed, the raw score is processed to a standard score and percentile that are age specific. The validity and reliability of this test varies between r = 0.57 and r = 0.91 [32].

#### 2.3.3. Beery–Buktenica Developmental Test of Visual–Motor Integration 4th Edition (VMI-4)

The Developmental Test of Visual–Motor Integration, fourth edition (VMI-4) [33] consists of three divisions, namely, visual–motor integration, visual perception and motor coordination. The aim of the VMI-4 is, by means of early identification, to identify children who require special help, and can be used for all age groups, from pre-school children to adults. The visual–motor integration division consists of three exercises and 24 progressively complex geometrical forms. The complete 27 item visual–motor integration division can be executed individually or in a group. It takes approximately 10 to 15 min to complete and is stopped when time has elapsed or after three consecutive mistakes have been made. The person is expected to copy a geometrical figure with a pencil without using an eraser; only one attempt is allowed per figure. The visual perception subtest requires the identification of corresponding forms and takes three minutes to complete or is stopped after the occurrence of three consecutive mistakes. The last subtest, motor coordination, involves the connecting of dots in a shape and takes five minutes to complete. The subtest is only stopped after the time has elapsed. The criteria for the VMI-4 scoring are as follows: a “0” is allocated for figures that are incorrect and a “1” is allocated to correct figures. The data are captured into three categories: visual–motor integration, visual perception and motor coordination. The raw score is converted to a standard score and then to a percentile. By using the standard score, children can be grouped into five different classes which vary from very high (133–160) to very low (40–67). The VMI-4 was developed to measure the extent to which the individual can integrate his or his visual and motor abilities. Poor results in the VMI-4 can be attributed to the inability to integrate visual and motor abilities and not necessarily to inadequate skills. The visual–motor integration and the complementary visual perception and motor coordination tests each have a general reliability of r = 0.92, r = 0.91 and r = 0.89, respectively [33].

### 2.4. Research Procedure

The Ethics Committee of the University, (NWU-0085-12-A1), gave ethical approval for this study. A meeting was arranged with the school principal during which the aim and protocol of the study were explained. Informed consent by the parents of each test subject was required before they could take part in the study. Each test subject also had to give accent after the protocol of the study was explained to them in detail. Participants whose parents gave consent for them to take part in the study, and gave accent themselves, were further evaluated with regards to their visual skills (ocular motor control, visual–motor integration, visual perception, motor coordination, hand–eye coordination, and reaction speed). The data collection took place during the second term of 2012 after which the intervention program started. The experimental group (*n* = 13) took part in a sports vision program for eight weeks, two sessions per week, each with a duration of 60 min. The control group (*n* = 11) did not partake in any sport vision or visual stimulation program during the afternoons; however, they still had to participate in the school’s netball practice session. After a period of eight weeks, the experimental group (*n* = 13) and the control group (*n* = 11) were tested to analyze the effect of the sports vision program. The experimental group (*n* = 13), as well as the control group (*n* = 11) underwent a retention test two months after the end of the sports vision program to determine the lasting effect of the sports vision program. The data collection was executed by trained Kinderkineticists, with a degree in Human Movement Sciences and postgraduate specialization in motor development, while presentation of the sports vision program was conducted by the researchers.

### 2.5. Intervention Program

A sports vision program was presented in groups as well as on an individual basis. The duration of the intervention program was 60 min, and was presented for eight weeks, twice a week during school hours on the selected school’s netball court. Visual exercises were combined with perceptual and gross motor activities. Motor activities (balance, hand–eye coordination, foot–eye coordination, bilateral integration and vestibular integration) focused on netball skills were combined with sport-specific visual exercises and were presented in groups that lasted approximately 50 min. The visual skills activities (peripheral vision, ocular motor control (visual tracking and fixation), depth perception, visual–motor integration, visual perception and motor coordination) were presented individually or in small groups and lasted approximately 10 min. The following apparatuses were used in the sports vision program: the Wayne Saccadic Fixator, Rotator Pegboard, Accommodation flippers (±1.00; ±1.50; ±2.00), the Red–Green Tranaglyph Orthoptics Kit, Free-space fusion cards, an eye patch, fixation objects and various Hart Cards (big and small) with respectively: individual letters, individual numbers, letters and numbers, colored dots and arrows. Additionally, an equilibrium board, pencils with numbers, letters and colors, balls, balance beams and a fine motor apparatus were used [6,34,35]. During the sports vision program, progressive work was carried out in three different but consecutive phases, namely: monocular (left eye and right eye are exercised separately), bi-ocular (both eyes are open, but the left eye does not see the same as the right eye); and lastly, binocular (both eyes are open and see exactly the same). Appendix A is an example of a progression that took place during the sports vision program.

### 2.6. Statistical Analysis

To process the data, SPSS 28.0 [36] was used. For descriptive purposes, the data were analyzed by means of arithmetic means (M), standard deviation (SD) and minimum and maximum values. To determine the effect of the intervention program within and between the control and experimental groups, use was made of mixed ANOVA (repeated measures over-time analysis of variance), as well as a Bonferroni correction for multiple comparisons. A *p*-value smaller than, or equal to, 0.05 was accepted as significant.

Effect sizes (d) were calculated to determine the practical significance of the results by dividing the mean difference of the two test opportunities by the greatest standard deviation (SD). For the interpretation of the practical significance, the following guidelines were used: d~0.2 indicates a small effect; d~0.5 indicates a medium effect and d~0.8 indicates a large effect [37]. As a result of the number of participants, it is seen as practically significant if the effect size (d-value) indicates a medium and/or larger effect.

## 3. Results

Table 1 provides descriptive information about the 24 participants who took part in the study and who were then divided into the experimental and control groups. The average age of the experimental group (*n* = 13) was 10.22 years (SD ± 0.65) while the control group’s age (*n* = 11) was a little lower with 9.97 years (SD ± 0.67)

Mixed model ANOVA with a Bonferroni adaptation was conducted to investigate the interaction between and within the control and experimental groups over time during the VMI-4, WFS and DEM subdivisions. Furthermore, box plots were used to indicate each participant’s values during the pre-, post and retention tests.

Table 2 indicates no statistically significant differences between the experimental and control group during the pre-test in any of the tests. The experimental group performed slightly worse than the control group in the VMI-4 as well as WSF subdivisions, even though these were not statistically significant. The results in Table 2 further indicated that during the visual–motor integration, visual perception and motor coordination no time*group interaction were reported. However, time did play a significant role (*p* ≤ 0.05) in the visual–motor integration and motor coordination test items, where an improvement in both groups was seen.

In Figure 2a and Figure 3a, it appears that the experimental group indicated a statistically (*p* ≤ 0.05) and practically (d ≥ 0.2) significant effect between the PrT and RT and the PoT and RT in the visual–motor integration. However, the control group also indicated a statistical (*p* ≤ 0.05) and practically (d ≥ 0.5) significant effect between the PrT and RT, while during the motor coordination subdivision, a statistically (*p* ≤ 0.05) and practically (d ≥ 0.5) significant effect occurred between the PrT and PoT and the PrT and RT (See Figure 3a–c). Furthermore, as seen in Figure 2a–c and Figure 3a–c, the control group outperformed the experimental group slightly in the VMI-4 sub-tests.

Regarding the WSF results, it is evident that a statistically significant interaction was evident between the two groups (*p* ≤ 0.05). The experimental group outperformed the control group in Procedure 1, 21 and 18 (see Figure 2d–f). The results furhter indicate a statistically (*p* ≤ 0.05) and a practically (d ≥ 0.5) significant effect with regards to all the sub-tests of the WSF with change that took place over time between the different test opportunities (between the PrT and PoT; between the PrT and the RT; between the PoT and the RT) in the experimental group respectively (see Figure 4a–c).

Lastly, regarding the DEM test, no time*group interaction was evident, but time did play a significant role (*p* = 0.009) during the vertical visual tracking, were the experimental group once again outperformed the control group (see Figure 2g,h and Figure 5a,b. The results indicate a statistically (*p* ≤ 0.05) and a practically (d ≥ 0.5) significant effect with regards to the two sub-tests of the DEM regarding the change that took place over time between the different test opportunities (between the PrT and PoT; between the PrT and the RT; between the PoT and the RT) in the experimental group, respectively.

## 4. Discussion

The aim of this study was to determine whether a sports vision program will improve the visual skills (ocular motor control, depth perception, visual–motor integration, visual perception, motor coordination and reaction speed) and hand–eye coordination of Grade 4 and 5 netball players.

From the results, it became apparent that the experimental group’s hand–eye coordination, peripheral vision and reaction speed improved statistically (*p* ≤ 0.05) upon following the sports vision program. In contrast to this, the control group weakened statistically (*p* ≤ 0.05) during the reaction speed subdivision. These results concur with those of various researchers [10,12,22,23,24,25] who have found that visual skills (visual tracking and peripheral vision), hand–eye coordination and reaction speed improved after following a sports vision program. The research by Du Toit et al. [24] found that after a sports vision program was followed (sessions of 10 min), the hand–eye coordination of 20 female rugby players between ages 19 to 24 improved, while the control group also showed improvement, though it was not statistically significant. In addition, the study by Du Toit et al. [28] carried out on 14- to 17-year-old boys who played rugby, found that these boys’ hand–eye coordination improved after following a sports vision program over five to eight weeks. Furthermore, the research findings by Krüger et al. [12] also indicate an improvement in the hand–eye coordination, peripheral awareness, and reaction speed of under-19 provincial cricket players after a sports vision program had been followed for eight weeks. The research findings by Abernethy and Wood [23] indicate an improvement in reaction speed after 40 participants with no experience in racket sports followed a sports vision program which was presented four times a week over a course of four weeks. In general accordance with our results and the above reported literature, Khanal [38] states that practitioners should realize the importance of vision training, as it is as important as physical training, if not more, to improve performance in sport.

The results further indicated that, although the experimental group showed improvement during the VMI-4 subdivisions after the sports vision program was presented, it was not statistically significant. Contrary to our results, Krüger et al. [12] found an improvement in the visual perception of under-19 provincial cricket players who were exposed to a sports vision program for eight weeks. A possible explanation for this phenomenon is that there was no adequate visual–motor integration, visual perception and motor coordination exercises in the sports vision program and that our participants were younger than the Krüger et al. [12] participants. Previous studies also report the specific value of perceptual learning approaches to improve sport-specific visual abilities and on-field performance [27]. Although we included perceptual abilities in our sport vision program, the focus was mainly on visual skills utilized in netball, which might have led to the improvements of non-significant nature seen in the VMI-4 sub-divisions.

During the DEM, the experimental group’s vertical and horizontal visual tracking improved statistically significantly (*p* ≤ 0.05) after the sports vision program had been followed. These results concur with the research findings of Du Toit et al. [24] and Krüger et al. [12], which indicate that after a sports vision program had been presented for five and eight weeks, the visual tracking skills of the participants improved significantly. Contrary to the results in this study, Abernethy and Wood [23] indicate no significant improvement in the visual skills of their participants after participation in a four-week sports vision program. Calder [7] is however of the opinion that there is a direct scientific relation between specific visual exercises and improved technical sports performance, and together with several other researchers, state that visual skills can be improved with a sports vision program [10,27]. With differences in visual skills and visuomotor abilities between genders, different age groups [21], and different situation/positional demands of each sport [9,21], sport vision programs should be tailored to improve visual skills most relevant to the specified sport situation. 

## 5. Conclusions

In conclusion, the results indicate that a sports vision program had a positive effect on the visual skills (visual tracking and peripheral awareness), hand–eye coordination and reaction speed of the participants who took part in the program. The results of our study must be judged in light of single shortcomings that were encountered in the study. This study consisted of a small group of participants (N = 24) and therefore generalization of the findings must be made with caution. Recommendations for future research include the execution of a more probing investigation in the field of the effect of sports vision programs on the visual skills of young (Grade 4 to 7) netball players. Longitudinal studies on a large sample of young athletes of various sports would also be advantageous to investigate the enhancement of visual performance by means of sport vision programs. Furthermore, it is also important to not only explore visual skill improvement by means of test battery results, but also to investigate the improvement in on-field visual skill performance after participation in a sport vision program, as both these aspects will influence overall performance on the field of play. Lastly, it is recommended for future studies to place focus on participants’ individual visual deficiencies. Despite the single shortcomings, the study brought forth valuable information with regards to the value that a sports vision program can have on Grade 4 and 5 netball players.

## Figures and Tables

**Figure 1 ijerph-19-09864-f001:**
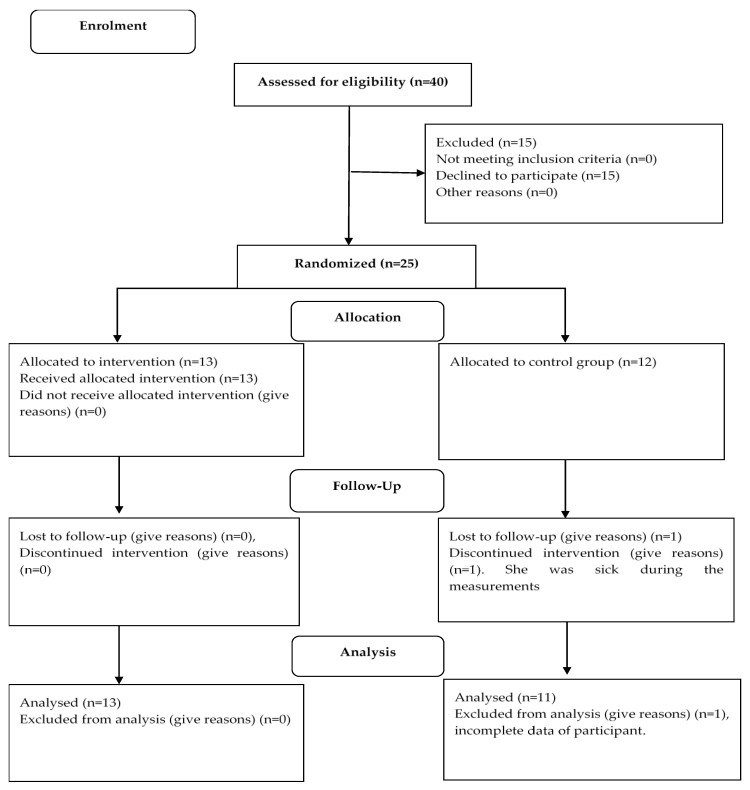
Flow diagram of the participants.

**Figure 2 ijerph-19-09864-f002:**
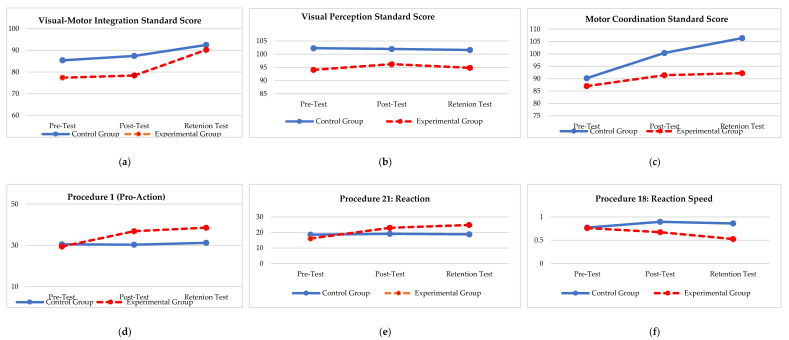
Group interaction over time regarding the Beery Visual–Motor Integration (VMI-4) test, Wayne Saccadic Fixator (WSF) and Developmental Eye Movement (DEM) test. (**a**) Group × Time effect for during pre-; post- and retention test (f = 3.33; *p* = 0.082). (**b**) Group × Time effect during pre-; post- and retention test (f = 1.58; *p* = 0.222). (**c**) Group × Time effect during pre-; post- and retention test (f = 5.18; *p* = 0.033). (**d**) Group × Time effect during pre-; post- and retention test (f = 12.34; *p* = 0.002). (**e**) Group × Time effect during pre-; post- and retention test (f = 3.69; *p* = 0.068). (**f**) Group × Time effect during pre-; post- and retention test (f = 26.02; *p* ≤ 0.001). (**g**) Group × Time effect during pre-; post- and retention test (f = 1.62; *p* = 0.217). (**h**) Group × Time effect during pre-; post- and retention test (f = 1.26; *p* = 0.275).

**Figure 3 ijerph-19-09864-f003:**
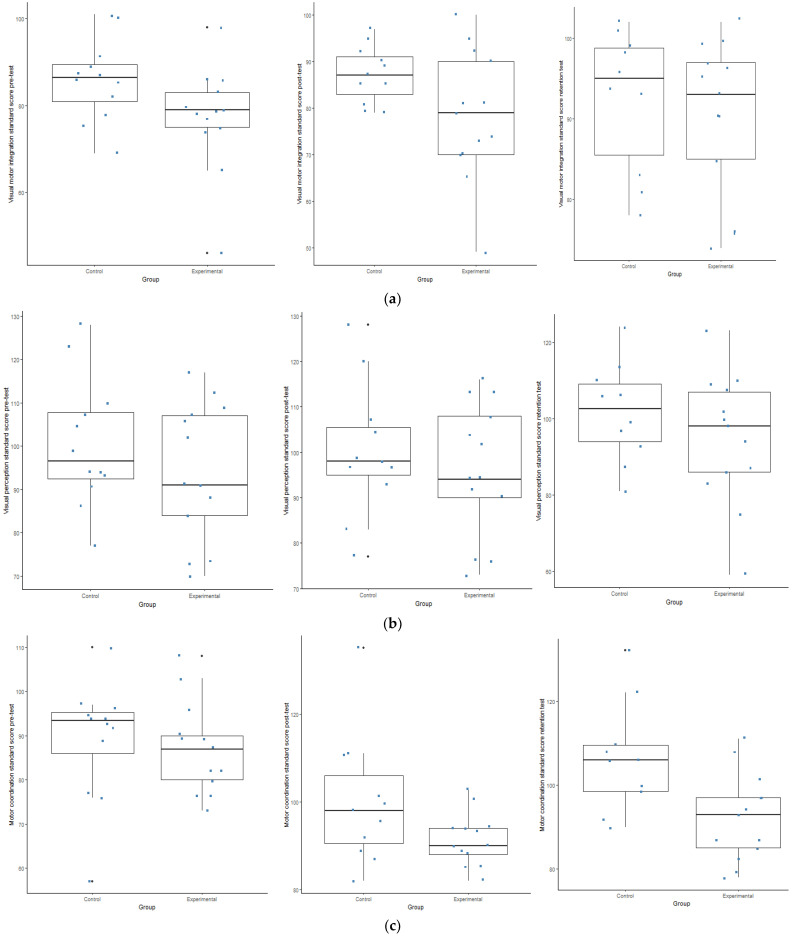
(**a**–**c**) Box plots indicate the interaction within the control and experimental groups during the Beery Visual–Motor Integration (VMI-4) test. (**a**) Box plots indicating the interaction within the groups during the pre-; post and retention test during the visual–motor integration test. (**b**) Box plots indicating the interaction within the groups during the pre-; post and retention test during the visual perception test. (**c**) Box plots indicating the interaction within the groups during the pre-; post and retention test during the motor coordination test.

**Figure 4 ijerph-19-09864-f004:**
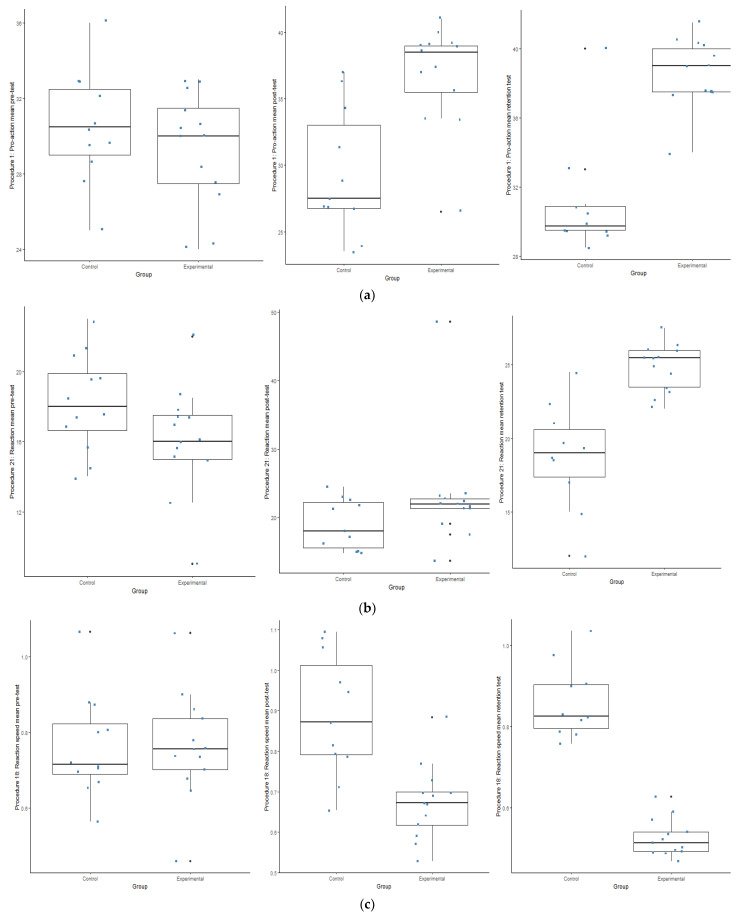
(**a**–**c**) Box plots indication the interaction within the control and experimental groups during the Wayne Saccadic Fixator (WSF) test. (**a**) Box plots indicating the interaction within the groups during the pre-; post and retention test during Procedure 1. (**b**) Box plots indicating the interaction within the groups during the pre-; post and retention test during Procedure 21. (**c**) Box plots indicating the interaction within the groups during the pre-; post and retention test during Procedure 18.

**Figure 5 ijerph-19-09864-f005:**
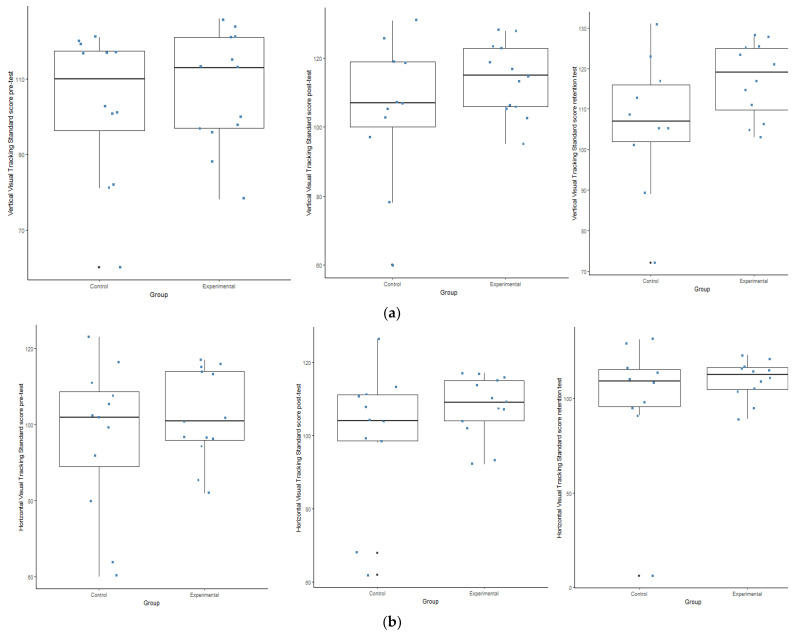
(**a**,**b**) Box plots indication the interaction within the control and experimental groups during the Development Eye Movement (DEM) test. (**a**) Box plots indicating the interaction within the groups during the pre-; post and retention test during the vertical visual tracking. (**b**) Box plots indicating the interaction within the groups during the pre-; post and retention test during the horizontal visual tracking.

**Table 1 ijerph-19-09864-t001:** Composition of experimental and control groups according to age.

Variables	Experimental Group (*n* = 13)	Control Group (*n* = 11)
Average age in years	10.22	9.97
Minimum age in years	9.50	9.10
Maximum age in years	11.30	11.00
Standard deviation (SD)	0.65	0.67

*n* = number of participants.

**Table 2 ijerph-19-09864-t002:** Effect of the sport vision program on the experimental and control group regarding the VMI-4, WSF and DEM test batteries.

Variable	df	Mean^2^	F-Value	*p*-Value
Visual–Motor Integration (VMI-4)				
Interaction effect	1	492,544.92	2335.20	≤0.001 *
Time effect	1.90	689.94	11.35	≤0.001 *
Time × Group effect	1.90	79.06	1.30	0.283
Group effect	1	701.79	3.33	0.082
Visual Perception (VMI-4)				
Interaction effect	1	658,185.57	1283.19	≤0.001 *
Time effect	1.73	7.21	0.07	0.912
TimexGroup effect	1.73	9.96	0.09	0.885
Group effect	1	810.90	1.58	0.222
Motor Coordination (VMI-4)				
Interaction effect	1	607,019.41	2396.27	≤0.001 *
Time effect	1.86	728.54	8.77	0.001 *
Time × Group effect	1.86	183.14	2.21	0.127
Group effect	1	1311.58	5.18	0.033 *
Procedure 1 (WSF)				
Interaction effect	1	68,729.14	2923.76	≤0.001 *
Time effect	1.95	142.97	25.14	≤0.001 *
TimexGroup effect	1.95	117.80	20.72	≤0.001 *
Group effect	1	290.02	12.34	0.002
Procedure 21 (WSF)				
Interaction effect	1	27,356.12	967.75	≤0.001 *
Time effect	1.42	182.16	8.02	0.004 *
Time × Group effect	1.42	156.73	6.90	0.007 *
Group effect	1	104.31	3.69	0.068
Procedure 18 (WSF)				
Interaction effect	1	38.00	1629.75	≤0.001 *
Time effect	1.98	0.05	6.14	0.005 *
Time × Group effect	1.98	0.16	18.04	≤0.001 *
Group effect	1	0.61	26.02	≤0.001 *
Vertical Visual Tracking (DEM)				
Interaction effect	1	769,758.38	1128.22	≤0.001 *
Time effect	1.39	492.71	6.59	0.009 *
Time × Group effect	1.39	80.64	1.08	0.332
Group effect	1	1106.26	1.62	0.217
Horizontal Visual Tracking (DEM)				
Interaction effect	1	689,061.601	751.560	≤0.001 *
Time effect	1.37	344.42	2.27	0.137
Time × Group effect	1.37	21.34	0.14	0.789
Group effect	1	1156.15	1.26	0.275

df—degrees of freedom; VMI-4—Beery test of Visual–Motor Integration; WFS—Wayne Saccadic Fixator; DEM—Development Eye Movement test; *p*—Greenhouse-Geisser; *p* ≤ 0.05 *.

## Data Availability

The dataset is the property of the North-West University under the supervision of Dané Coetzee. In this regard D. Coetzee should be contacted if, for any reason, the data included in this paper needs to be shared. D. Coetzee is the principal investigator of this study and gave permission that the data can be used.

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
