# Peer review of "An Exploratory Investigation of the Effect of a Sports Vision Program on Grade 4 and 5 Female Netball Players’ Visual Skills"

_ijerph, 2022, doi:10.3390/ijerph19169864_

Round 1
Reviewer 1 Report
See attached file.

Author Response
Thank you very much for the time that you took to review our article. We value your input to make this article even stronger. Please find attached our rebuttal letter.
Kind regards

Reviewer 2 Report
Please show boxplots of the results with individual data points corresponding to each tested subject. Visualizing the data like this would give a better idea of the training program's impact on each subject.
Is there a scientific reason why this study did not include boys?
Have you noticed any specific improvement in netball skills?
In this study the authors investigate whether a sports vision program has the potential of benefitting the visual skills and hand-eye coordination of Grade 4 and 5 female netball players. The authors recruited 25 girls which were separated into an experimental group that underwent the sports vision program and a control group which did not undergo the program. The skills assessed are hand-eye coordination, reaction times, peripheral vision, oculomotor function and visuo-motor integration. Each group underwent a pre-test, post-test and retention test.
Significant statistical differences are found between the control and experimental groups after the intervention in procedures 1 and 18 of the WFS, which remain during the retention test. The intervention program also seemed to influence motor coordination.
The topic is not original, however more systematic investigations into how different aspects of visual perception affect sports performance should be performed. This field lacks such studies that use modern visual perception investigation techniques to perform such rigorous investigations
This study is relevant as is contributes to the effort of increasing knowledge in that field.
The added value of this study is that it investigates visual skill in a new sport: net ball, and in a not so investigated age group. The scientific value is also slightly better, although many things can be improved.
The authors use standardized tests to evaluate visual function. However, many variables that the authors did not specify can influence the outcome:
· The authors failed to mention the visual health condition of the participants, whther they had corrected vision, whether they had strabismus, or under conditions.
· The authors also did not specify the inclusion criteria
· The authors must include some additional information such the distance at which participants stood from the Wayne Fixator, the hand they use to respond, the dominant eye of participants, and their height.
The authors also aimed to evaluate saccadic eye movements. However their method is only an indirect measure of the properties of eye movements, as an eye-tracking device is necessary to do so.
The authors should also improve how they present and statistically analyze data. Data presentation should be done using boxplots showing the median, std and 95% confidence intervals, as well as individual data points. The authors should also use a mixed-model anova to compare the results within and between groups.
The authors make reasonable conclusions regarding their findings, however the mixed model anova should use to corroborate their conclusions.
The references are appropriate.
The tables in the results are redundant. Results should be present at boxplots as mentioned above. A figure showing the wayne fixator and some exapmles of the tests used would greatly help comprehensions of the reader.
Author Response

(The authors gave the same response as above.)
